# Improving EFDD with Neural Networks in Damping Identification for Structural Health Monitoring

**DOI:** 10.3390/s25226929

**Published:** 2025-11-13

**Authors:** Yuanqi Zheng, Chin-Long Lee, Jia Guo, Renjie Shen, Feifei Sun, Jiaqi Yang, Alejandro Saenz Calad

**Affiliations:** 1Department of Civil and Environmental Engineering, University of Canterbury, Christchurch 8041, New Zealand; chin-long.lee@canterbury.ac.nz (C.-L.L.); alejandro.saenzcalad@pg.canterbury.ac.nz (A.S.C.); 2Division of Environmental Science and Technology, Kyoto University, Kyoto 606-8502, Japan; guo.jia.8x@kyoto-u.ac.jp; 3Department of Engineering Mechanics, School of Naval Architecture & Ocean Engineering, Jiangsu University of Science and Technology, Zhenjiang 212003, China; rjshen@just.edu.cn; 4College of Civil Engineering, Tongji University, Shanghai 200092, China; ffsun@tongji.edu.cn (F.S.);; 5State Key Laboratory of Disaster Reduction in Civil Engineering, Tongji University, Shanghai 200092, China

**Keywords:** operational modal analysis, enhanced frequency domain decomposition, damping identification, heuristic parameter, multilayer perceptron, damage detection

## Abstract

Damping has attracted increasing attention as an indicator for structural health monitoring (SHM), owing to its sensitivity to subtle damage that may not be reflected in natural frequencies. However, the practical application of damping-based SHM remains limited by the accuracy and robustness of damping identification methods. Enhanced Frequency Domain Decomposition (EFDD), a widely used operational modal analysis technique, offers efficiency and user-friendliness, but suffers from intrinsic deficiencies in damping identification due to bias introduced at several signal-processing stages. This study proposes to improve EFDD by integrating neural networks, replacing heuristic parameter choices with data-driven modules. Two strategies are explored: a step-wise embedding of neural modules into the EFDD workflow, and an end-to-end grid-weight framework that aggregates candidate damping estimates using a lightweight multilayer perceptron. Both approaches are validated through numerical simulations on synthetic response datasets. Their applicability was further validated through shaking-table experiments on an eight-storey steel frame and a five-storey steel–concrete hybrid structure. The proposed grid-weight EFDD demonstrated superior robustness and sensitivity in capturing early-stage damping variations, confirming its potential for practical SHM applications. The findings also revealed that the effectiveness of damping-based indicators is strongly influenced by the structural material system. This study highlights the feasibility of integrating neural network training into EFDD to replace human heuristics, thereby improving the reliability and interpretability of damping-based damage detection.

## 1. Introduction

Structural damage detection has long been a central topic in civil engineering. From a technical perspective, the available methods can be broadly classified into non-destructive testing (NDT) and vibration-based approaches [1,2]. NDT methods are local, requiring prior knowledge of the damaged region and direct accessibility for inspection [3,4,5], which cannot be guaranteed when the structure is large or its operating environment is complex. Vibration-based approaches infer structural condition by monitoring changes in global dynamic properties such as natural frequencies, mode shapes, and modal damping [6,7,8,9,10]. This category is commonly referred to as Structural Health Monitoring (SHM).

As indicators for SHM, natural frequencies are relatively insensitive to damage severity [11], while mode shapes require a dense sensor network to be effectively identified [1]. Modal damping has attracted increasing attention as a potential indicator in recent research. Numerous experimental and field studies have reported that damping ratios exhibit noticeable changes under different damage scenarios [12,13,14,15,16,17,18,19,20]. Previous studies consistently indicate that damping ratios are indeed more sensitive than natural frequencies as indicators for SHM. Even small cracks in concrete or minor corrosion in reinforcing steel can cause changes in damping several to tens of times larger than the corresponding changes in frequency. Nevertheless, some investigations [21,22,23] have concluded that damping is a rather unreliable damage indicator, as the identified values did not exhibit obvious or consistent trends. These discrepancies can be attributed to two main factors.

Excitation conditions differ across studies. The physical relation between the damping coefficient and damage remains insufficiently clear owing to the complexity of damping mechanisms. In practice, viscous damping, hysteretic damping, and frictional damping may act simultaneously. When identification is performed under the assumption of viscous damping, the resulting estimates can be interpreted as equivalent viscous damping, which indicates the energy dissipation rate during vibration. Different types of damage may lead to contrasting trends in damping variation, and the mechanisms involved under forced vibration are generally more complex than those under free decay.Measurement noise exerts a significant influence. When damping values are small, the immaturity of estimation techniques and operational errors can strongly affect the accuracy of damage detection. Achieving reliable damping identification under forced vibration conditions remains a pressing problem to be addressed [24,25].

To address these challenges, this study adopts Enhanced Frequency Domain Decomposition (EFDD) [26] as the base framework and integrates neural networks to enhance the accuracy of damping ratio identification. EFDD is a widely used operational modal analysis (OMA) technique [27]. OMA allows economical and rapid testing without interrupting the normal operation of structures. EFDD is considered user-friendly and computationally efficient. It is broadly applicable across various structural materials and types, as it exploits output-only vibration data and does not rely on specific constitutive models. It has equal relevance to masonry, timber, and reinforced-concrete buildings, and can be directly extended to bridges. As with other modal-identification techniques, validity is ensured when the response is approximately linear and the excitation provides sufficient bandwidth. However, the method remains limited in identifying damping ratios. It estimates modal damping ratios by transforming the frequency-domain singular value spectrum into the time domain and applying logarithmic decrement regression. Biases arise at multiple stages of the EFDD procedure [28,29,30,31,32,33]. Recent studies have begun to incorporate neural networks into FDD: LSTM-assisted FDD for stabilising modal analysis [34], hybrid NN-based OMA under non-white excitation [35], and PSD-driven GNN frameworks for automated modal identification across structural populations [36]. These efforts primarily address robustness or automation of modal extraction and do not explicitly resolve EFDD’s multi-stage bias for damping identification. This study examines the integration of neural networks into the EFDD framework to mitigate these biases. The detailed formulations and training methodologies are presented in the following sections.

## 2. Bias of EFDD and Neural Network Integration

EFDD begins with the estimation of the cross-power spectral density (PSD) matrix $S_{yy}\left(f\right)$ from multi-channel responses y(t) using Welch’s approach. At each frequency line *f*, a singular value decomposition (SVD) is performed: (1)$$S_{yy}\left(f\right)=U\left(f\right)\;\Sigma \left(f\right)\;U^{H}\left(f\right)$$
where Σ(f) contains the singular values $\lambda_{1}\left(f\right)\geq \lambda_{2}\left(f\right)\geq \ldots$. When a mode dominates, the first singular value $\lambda_{1}\left(f\right)$ exhibits a clear spectral peak. EFDD proceeds by selecting a frequency band around the peak, applying an inverse FFT to obtain the free-decay function r(t), and then fitting the log-envelope(2)$$y_{t}=\ln\left|r\left(t\right)\right|\approx a-\delta t$$
to estimate the decay constant δ and damping ratio $\zeta =\delta /(2\pi f_{n})$. The fitting of the free-decay function r(t) also provides the corresponding natural frequency $f_{n}$ of mode *n*. This process can be interpreted as a refinement of the modal frequency identified in the initial peak-picking stage. Further details can be found in the previous studies [26,33,37,38,39]. EFDD damping identification is highly sensitive to subjective choices at three stages [28,29,30,31,32,33]:The selection of Welch parameters for power spectral density estimation, which introduces a trade-off between variance and leakage.The choice of frequency bands for modal isolation may result in either mode contamination or information loss.The truncation and regression of the log-envelope, which are highly sensitive to noise and gating heuristics.

Improper decisions in these steps introduce bias and variance, leading to inconsistent damping estimates. Traditionally, users address these issues through manual tuning. For Welch parameters, engineers vary the segment length, overlap, and window type until the dominant peaks appear sufficiently distinct. For band selection, modal assurance criterion (MAC) thresholds or visual inspection are used to delimit the frequency range associated with each mode. For time gating, fixed rules such as ‘skip three peaks and fit the following 10 to 15 cycles’ are applied to mitigate transient and noise effects. While these remedies have enabled EFDD to become a practical tool, they remain highly subjective and user-intensive. This motivates replacing heuristic rules with data-driven neural network (NN) modules that can consistently approximate the underlying optimisation objectives.

The present study introduces three dedicated neural modules. A multilayer perceptron (MLP) [40] is employed to learn global statistical mappings for Welch parameter selection, a convolutional neural network (CNN) [41] is used to capture localised frequency-domain patterns for band determination, and a long short-term memory (LSTM) [42] network is designed to model temporal dependencies for envelope gating. These architectures are selected according to their respective strengths in regression [43,44,45], feature extraction [46,47,48], and sequential learning [49,50,51]. The following subsections reformulate each step mathematically and describe how they are integrated to enhance EFDD.

### 2.1. Welch Parameters

For a segment of the signal $x_{k}\left[n\right]$ of length *N*, the Welch PSD [52] is estimated as(3)$${\hat{S}}_{xx}\left(f\right)=\frac{1}{KU}\sum_{k=1}^{K}{\left|\sum_{n=0}^{N-1}w\left[n\right]\;x_{k}\left[n\right]\;e^{-j2\pi fn}\right|}^{2},\;U=\frac{1}{N}\sum_{n=0}^{N-1}w^{2}\left[n\right]$$
where *K* is the number of segments, w[n] is the window function (Hann, Hamming, etc.), *U* is the window normalisation constant, and *N* gives frequency resolution $df=f_{s}/N$.

Its variance scales with the effective number of averages $K_{\mathrm{eff}}$: (4)$$\mathrm{var}\left({\hat{S}}_{xx}\left(f\right)\right)\approx \frac{2}{\nu }\;S_{xx}^{2}\left(f\right),\;\nu \approx 2K_{\mathrm{eff}}$$
where $K_{\mathrm{eff}}$ is smaller than the nominal number of segments *K* whenever overlap is applied, because overlapping segments are statistically correlated. The exact reduction factor depends on the overlap percentage and the window shape w[n].

The expectation of PSD is biased due to spectral leakage: (5)$$E\left[{\hat{S}}_{xx}\left(f\right)\right]={\left(|W\right|}^{2}*S_{xx})\left(f\right).$$

Leakage mixes energy from neighbouring modes, reducing the separation $\lambda_{1}/\lambda_{2}$ and corrupting modal identification. Hence, the Welch parameter vector is given as θ=(df, overlap, window). The selection of these parameters can be formulated as an optimisation problem [53,54]: (6)$$\theta^{★}=\arg\;\max_{\theta }\;\left(\underset{\mathrm{mode}\;\mathrm{separation}}{\underbrace{\int_{N(f_{0})}\log\;\frac{\lambda_{1}\left(f\right)}{\lambda_{2}\left(f\right)}\;df}}-\beta \underset{\mathrm{variance}\;\mathrm{penalty}}{\underbrace{\int \frac{\mathrm{var}(\hat{S}\left(f\right))}{S^{2}\left(f\right)}\;df}}-\gamma \underset{\mathrm{leakage}\;\mathrm{penalty}}{\underbrace{\int \left(|W_{\theta }{|}^{2}*S\right)\left(f\right)\;df}}\right)$$
where $N(f_{0})$ is frequency neighborhood around the target modal frequency $f_{0}$, S(f) is the true spectrum at frequency *f*, $|W_{\theta }{|}^{2}$ is the power spectrum of the window function determined. The weighting factors β and γ in Equation (Equation 6) are introduced to balance terms of different scales.

In practice, direct optimisation of Equation (Equation 6) for each response is computationally expensive and requires access to the true spectrum S(f), which is unavailable in experimental settings. Therefore, a surrogate learning strategy is adopted. A set of candidate parameter configurations $\left\{\theta_{i}\right\}$ is first evaluated on the training responses using the classical EFDD procedure. Each configuration yields a corresponding damping estimate $\zeta_{i}$ and a spectrum-derived quality score $q_{i}$. For each candidate $\theta_{i}$, a normalised quality score is calculated as(7)$$q_{i}=\alpha \;\underset{\mathrm{peak}\;\mathrm{prominence}}{\underbrace{\mathrm{Prom}\;\left(\lambda_{1};N\left(f_{0}\right)\right)}}+\beta \;\underset{\mathrm{SV}\;\mathrm{separation}}{\underbrace{\mathrm{Sep}\;\left(\frac{\lambda_{1}}{\lambda_{2}};N\left(f_{0}\right)\right)}}-\gamma \;\underset{\mathrm{spectral}\;\mathrm{roughness}}{\underbrace{\mathrm{Rgh}\;\left(\lambda_{1};N\left(f_{0}\right)\right)}},$$
where Prom is the median-subtracted peak height of $\lambda_{1}\left(f\right)$ within $N(f_{0})$, Sep is the mean of $\log(\lambda_{1}/\lambda_{2})$ over $N(f_{0})$, and Rgh is the $L_{2}$-norm of the first finite difference of $\lambda_{1}\left(f\right)$. The weights (α,β,γ) are set on a held-out development set (defaults: 0.4,0.4,0.2).

The configuration with the highest score, $\theta^{\mathrm{pseudo}}=\arg\max_{\theta_{i}}q_{i},$ is regarded as the pseudo-optimal label that implicitly satisfies the balance in Equation (Equation 6). These pseudo-labels form a training set $\left\{\left(\phi^{\left(n\right)},\;\theta^{\mathrm{pseudo},(n)}\right)\right\}$, where $\phi^{\left(n\right)}$ denotes the global spectral statistics extracted from the corresponding response.

A lightweight MLP, termed *WelchHead*, is trained to approximate the mapping $F_{W}:\phi \mapsto \theta$ by minimising a supervised regression loss: (8)$$L_{W}=\sum_{n}∥F_{W}\left(\phi^{\left(n\right)}\right)-\theta^{\mathrm{pseudo},(n)}{∥}_{2}^{2}.$$
Through this pseudo-label imitation framework, the network learns to reproduce the implicit trade-off among mode separation, variance, and leakage penalties without explicitly computing the objective in Equation (Equation 6). As a result, the WelchHead is trained to predict the optimal parameter set θ=(df,overlap,window) based on spectral statistics $\phi (\hat{S})$ such as peak-to-median ratios, variance, and mean separation $\lambda_{1}/\lambda_{2}$. The workflow is shown in Figure 1. Through the pseudo-label imitation process, the network implicitly learns how to balance the competing objectives of Equation (Equation 6), thereby replacing the manual tuning of β and γ with a data-driven mapping from global spectral features to well-balanced Welch parameters.

### 2.2. Band Selection

Near a modal frequency $f_{0}$, the dominant spectral line is approximated as(9)$$S_{1}\left(f\right)=\frac{A}{{\left(f-f_{0}\right)}^{2}+{\left(\zeta f_{0}\right)}^{2}}.$$

Let $M(f;\;f_{L},\;f_{U})$ be the rectangular band-pass indicator of $B=[f_{L},f_{U}]$. The band-limited inverse transform then gives(10)$$r\left(t;B\right)\propto \int_{f_{L}}^{f_{U}}S_{1}\left(f\right)\;e^{j2\pi ft}\;df\approx Ce^{-\delta t}\cos\left(2\pi f_{d}t\right)\cdot \mathrm{sinc}\left(t;|B|\right).$$
If $B=[f_{L},f_{U}]$ is too narrow, sinc modulation introduces oscillations. If |B| is too wide, the band captures undesired energy from neighbouring modes and produces beating that bends the log-envelope. The band quality is quantified by $R^{2}\left(B\right)$: linearity, C(B): curvature penalty, M(B): monotonicity penalty, ε(B): mode contamination, A(B): asymmetry, W(B): bandwidth regularization. For clarity, the mathematical definitions of these band-quality metrics on a selected time index set T are not repeated here. Their derivation follows directly from standard signal processing principles. Readers can refer to Jacobsen et al. [31] and subsequent EFDD reviews [32,38] for detailed formulas and discussions.

The optimisation of band selection can be defined as Equation (Equation 11) [26,38] with nonnegative weights α,λ,η,ρ,κ.(11)$$B^{★}=\arg\;\max_{f_{L}<f_{U}}\;R^{2}\left(B\right)-\alpha C\left(B\right)-\lambda M\left(B\right)-\eta \varepsilon \left(B\right)-\rho A\left(B\right)-\kappa W\left(B\right).$$

In practice, the CNN-based BandHead does not directly optimise Equation (Equation 11) by gradient descent. Instead, it is trained via a surrogate imitation process. For each training response, multiple candidate bands are evaluated using the classical EFDD procedure and assigned a quality score reflecting envelope linearity, monotonicity, and spectral isolation. The band with the highest score, $B^{\mathrm{pseudo}}=\left[f_{L}^{★},f_{U}^{★}\right]$, is treated as the pseudo-optimal label. The CNN is then supervised to regress $\left(\Delta f_{L},\Delta f_{U}\right)$ that best reproduce these pseudo-labels: (12)$$L_{\mathrm{Band}}=∥\hat{\Delta f_{L}}-\Delta f_{L}^{★}{∥}_{2}^{2}+∥\hat{\Delta f_{U}}-\Delta f_{U}^{★}{∥}_{2}^{2}.$$
Through this pseudo-label imitation, the network implicitly learns the underlying trade-offs among curvature, monotonicity, contamination, and bandwidth penalties in Equation (Equation 11). Consequently, the BandHead achieves consistent, data-driven band selection across varying spectra while avoiding the need for manual tuning of the weighting factors (α,λ,η,ρ,κ).

### 2.3. Time Gating and Robust Regression

After band selection, the band-limited inverse transform r(t;B) yields the free-decay signal whose log-envelope ideally follows a straight line: (13)$$y_{t}=\ln\left|r\left(t;B\right)\right|\approx a-\delta t,$$
where *a* is the intercept and $\delta =2\pi \zeta f_{n}$ is the decay constant associated with damping ratio. In practice, however, the observed envelope deviates due to transient effects, noise floor, and possible beating from residual modes: (14)$$y_{t}=a-\delta t+b\left(t\right)+\varepsilon_{t},$$
where b(t) is a low-frequency bias term (e.g. beating) and $\varepsilon_{t}$ is stochastic noise. Least-squares regression over all samples introduces significant bias because early points (windowing transients) and late points (noise floor) dominate the fit.

To address this, the Huber loss is adopted to mitigate the influence of outliers: (15)$$\min_{a,\delta }\sum_{t}w_{t}\;\rho \left(y_{t}-\left(a-\delta t\right)\right),\;\rho \left(u\right)=\left\{\begin{array}{cc} \frac{1}{2}u^{2}, & |u|\leq \Delta ,\\ \Delta (|u|-\frac{1}{2}\Delta ), & |u|>\Delta , \end{array}\right.$$
where $w_{t}=g\left(t\right)/\bar{g}$ are normalised gating weights with g(t)∈[0,1]. To regularise the gate, physical priors are imposed to penalise upward slopes and excessive curvature in the log-envelope [55]: (16)$$\min_{a,\delta ,g}\;\sum_{t}g_{t}\rho \left(y_{t}-\left(a-\delta t\right)\right)+\lambda_{1}\sum_{t}g_{t}{\left[y_{t}^{\prime }\right]}_{+}^{2}+\lambda_{2}\sum_{t}g_{t}{\left(y_{t}^{\prime \prime }\right)}^{2},$$
where ${\left[y_{t}^{\prime }\right]}_{+}=\max\left(y_{t}-y_{t-1},0\right)$ and $y_{t}^{\prime \prime }=y_{t+1}-2y_{t}+y_{t-1}$.

A bi-directional long short-term memory (Bi-LSTM) network, termed *GateHead*, is employed to infer the gating weights in a data-driven manner. The input to the network is a sequence of envelope-based features, $\Psi_{t}\in \left\{y_{t},\;\Delta y_{t},\;\Delta^{2}y_{t},\mathrm{moving}\;\mathrm{averages},\;\mathrm{local}\;\mathrm{variances},\;\;\ldots \right\}$ which capture local slope, curvature, and smoothness of the decay curve. The network outputs a continuous weighting profile ${\left\{g\left(t\right)\right\}}_{t=1}^{T}$ that highlights the physically meaningful decay segment for damping regression. A weighted robust regression using these predicted gates then yields the decay constant δ, from which the damping ratio ζ is obtained.

Since the true optimal gating functions are not available for real data, the GateHead is trained on synthetic or preprocessed free-decay signals with known damping. For each training instance, the optimal gating mask $g^{★}\left(t\right)$ and decay constant $\delta^{★}$ are obtained through offline minimisation of Equation (Equation 16). The LSTM is then trained to imitate this mapping by minimising(17)$$L_{\mathrm{Gate}}=∥g\left(t\right)-g^{★}{\left(t\right)∥}_{2}^{2}+\gamma \left|\hat{\delta }-\delta^{★}\right|,$$
where the first term enforces consistency with the reference gate and the second maintains the regression accuracy of the estimated decay constant. Through this supervised imitation process, GateHead implicitly learns the functional behaviour of Equation (Equation 16), replacing the empirical rule-based procedures (e.g., “skip three peaks and fit ten cycles”) with a smooth, data-adaptive gating mechanism that generalises well across noise levels and datasets.

Table 1 summarises the three neural modules developed in this study. All neural heads are trained with Adam (initial learning rate $10^{-3}$), weight decay $10^{-4}$, and early stopping based on validation loss (patience of 20 epochs). For WelchHead and BandHead, a top-*k* neighbourhood micro-check (with k=3) is conducted around the predicted setting. Predictions are accepted only if the refined envelope coefficient of determination exceeds a threshold (default $R^{2}\geq 0.92$).

## 3. NN-Embedding Framework

Building upon the mathematical overview in the previous section, this part focuses on strategies for embedding neural networks into the EFDD framework. Two main approaches are considered: step-wise and grid-weight.

For step-wise, three neural modules (WelchHead, BandHead, and GateHead) are introduced as plug-ins to assist parameter selection, as shown in Figure 2a. While this modular strategy provides interpretability and preserves EFDD’s mathematical consistency, it also entails certain limitations. The optimisation objectives are human-crafted surrogates, reflecting idealised bias–variance trade-offs rather than the true end goal of damping identification. Consequently, the neural modules learn to reproduce these surrogate losses. The ultimate performance metric, the error between the estimated and the true damping ratio, does not directly propagate back to guide their training. It risks accumulating approximation errors at each stage and decouples local optimisation from the global target.

A direct grid-weight approach is proposed to overcome these drawbacks. Instead of replacing EFDD steps with surrogate optimisation, EFDD is run in its conventional form, but with systematic parameter sweeps at the bias-sensitive stages. For instance, multiple combinations of Welch parameters, bands, or gating rules are explored, producing a multidimensional grid of damping ratio candidates. These trial results are then combined by training a lightweight MLP as a grid weight aggregator. The network takes as input both global spectral statistics and local frequency-domain features, and as outputs softmax weights that form a weighted average over the candidate damping ratios.

This approach transforms EFDD into an end-to-end trainable system, as shown in Figure 2b. The model is trained directly to minimise the error between the weighted damping ratio and the ground truth, thereby ensuring that the final performance metric is aligned with the training objective. The advantages are threefold: 1. The robustness of EFDD is retained, since all candidate ratios stem from physically interpretable steps. 2. The learning burden on the MLP is lightweight, focusing only on weighting rather than reproducing the entire identification process. 3. End-to-end supervision ensures that error feedback is directly linked to the damping ratio estimate, eliminating the detour through surrogate objectives.

In summary, both neural embedding strategies share a common objective: to replace heuristic rules in EFDD with data-driven learning mechanisms that mitigate bias across the three critical stages. While the step-wise approach emphasises interpretability and alignment with EFDD’s analytical structure, the grid-weight approach achieves global optimisation by directly coupling the learning objective with the final damping accuracy.

## 4. Numerical Verification

A three-degree-of-freedom (3-DOF) shear-type linear structure is built, as shown in Figure 3, where m=363t, $k=1.07\times 10^{5}\;\mathrm{kN}/m$. All the responses are simulated using the Newmark constant acceleration method (β=0.25,γ=0.5) with a time step Δt=1/256s. The original natural frequencies are 1, 2.73 and 3.73 Hz. Modal damping is modelled using a classical Wilson–Penzien formulation with a constant damping ratio assigned to all modes. Although Wilson–Penzien is sufficient for linear simulations, it may not well represent damping behaviour in nonlinear or inelastic structures. In such cases, bell-shaped damping models have been demonstrated to provide more physically consistent energy dissipation representation during time–history analyses [56,57,58,59], and readers are encouraged to consider their use for inelastic response simulations.

A set of 300 s white noise responses is generated by adjusting the model’s mass, giving various natural frequencies of the structure from 1 Hz to 5 Hz in increments of 1 Hz. In practical applications of EFDD, very long records are often used for damping identification, sometimes extending to 30 min or more [27,34,60]. Hence, estimating damping ratios from a 300 s response poses significant challenges. In this study, the duration and sampling frequency were chosen to remain consistent with the subsequent experimental data.

The damping ratios of the three modes were varied from 1% to 8% in increments of 0.1%. A total of 355 responses were simulated. All numerical responses were superimposed with Gaussian white noise at a signal-to-noise ratio (SNR) of 35 dB to represent measurement noise [61]. The simulated response datasets were used as training data following the flow outlined in Section 3.

In addition, an independent set of 355 homogenised responses was generated using the same procedure and reserved exclusively for testing, without being involved in model training. Traditional EFDD was applied for baseline comparison with its default parameter settings [62] (frequency resolution 0.01 Hz, overlap 0.5, Hann window, MAC threshold 0.95, skipping the first three peaks and fitting the subsequent 15 cycles). The effectiveness of the proposed neural modules was evaluated through Monte Carlo simulations against this baseline.

### 4.1. Step-Wise

For WelchHead, parameters were optimised using a multilayer perceptron (MLP). The training targets were obtained by a grid search over segment resolutions {0.005,0.01,0.02} Hz, overlaps {0.3,0.5,0.7}, and window types {Hann, Hamming, Blackman}. From each trial, eight global spectral statistics were extracted, including peak-to-median ratio, variance, and singular-value separation, which served as the input features. The MLP was trained for 1000 epochs with a batch size of 8, using mean-squared error for resolution and overlap, and cross-entropy loss for window type.

For BandHead, a one-dimensional CNN was combined with a shallow feedforward layer. Neighbourhood features were extracted from a ±1.0 Hz window around the target frequency, consisting of seven channels (first and second singular values in dB, their ratio, first- and second-order derivatives, moving averages, and local variances). Candidate bands were generated from offsets $\Delta f_{L}\in \left[-0.8,-0.2\right]$ Hz and $\Delta f_{U}\in \left[0.2,0.8\right]$ Hz, with pseudo-labels obtained from the band yielding the most accurate damping estimation. The network was trained for 1000 epochs. A mode-dependent weighting scheme was applied in the loss aggregation, with weights of 0.7, 0.2, and 0.1 for the first, second, and third modes, respectively, reflecting their relative importance.

For GateHead, a bidirectional LSTM was trained to predict a soft selection mask for the log-envelope. The input sequence contained three channels: the log-envelope itself, its first derivative, and its second derivative. The loss function combined binary cross-entropy with monotonicity and smoothness penalties, encouraging the selected envelope to be stable and physically interpretable. Training was conducted for 1000 epochs with a batch size of 8, again with mode-dependent weights of 0.7, 0.2, and 0.1.

All models were trained with the Adam optimiser at a $10^{-3}$ learning rate and weight decay of $10^{-4}$. To ensure comparability, damping estimates were projected onto the non-negative domain. Figure 4 presents the loss functions obtained during the training of the step-wise neural modules. For WelchHead and GateHead, the loss decreases steadily as the number of epochs increases, indicating effective convergence of the models. In contrast, the loss curve of BandHead exhibits an abrupt drop at the beginning of training but fails to show a clear downward trend afterwards, suggesting the limited effectiveness of the learning process and insufficient alignment with the optimisation objective.

When applied to the 355 test responses, the identification errors were used to construct error distribution plots. The results indicated that the overall accuracy of damping estimation degraded when compared with the traditional EFDD baseline. Consistent with the observations in Figure 4, where BandHead exhibited poor convergence behaviour, this module was discarded, and the models incorporating only WelchHead and GateHead were re-evaluated. Figure 5 and Figure 6 present the corresponding identification results for natural frequencies and damping ratios, respectively. The training process did not explicitly target the accuracy of frequency identification. Therefore, the results in Figure 5 are provided to confirm that the step-wise models do not compromise frequency estimates.

Figure 5 shows that when all three step-wise NN modules were included, the identification accuracy of the first and second modal frequencies exhibited a slight reduction, while that of the third mode deteriorated considerably. By contrast, the model excluding BandHead demonstrated modest improvements across all three modes. Overall, the errors remained within an acceptable range, confirming the inherent stability of EFDD in frequency identification.

Figure 6 demonstrates that the model combining WelchHead and GateHead provides a clear advantage in identifying the damping ratio of the first mode, while the results for the second and third modes are comparable to the baseline. In contrast, the full three-step step-wise model exhibited a pronounced degradation in the accuracy of the first-mode damping ratio, but yielded noticeable improvement in the identification of the third mode. In summary, the step-wise model was finalised using the WelchHead and GateHead modules, while the BandHead component was excluded due to its limited effectiveness.

### 4.2. Grid-Weight

The proposed grid-weight approach aims to replace heuristic parameter selection in EFDD by learning soft weights over a precomputed grid of candidate damping ratios. The training procedure is outlined in Section 3 and Figure 2b.

For each simulated response, EFDD is executed on a three-dimensional grid of parameters. Hanning, Hamming and Blackman window types are used, and the

MAC threshold ranges from 85% to 97%. Time gating skips three peaks and fits the following cycles from 8 to 20. This yields 3×13×13=507 candidate damping estimates. The candidate vector for a given mode is denoted as $\zeta^{\mathrm{cand}}={\left[\zeta_{1}^{\mathrm{cand}},\zeta_{2}^{\mathrm{cand}},\ldots ,\zeta_{507}^{\mathrm{cand}}\right]}^{⊤}.$

Two groups of features are extracted from each response as input. One is global spectral statistics $\phi_{g}\in R^{8}$, such as peak-to-median ratio, variance, and singular-value separation, computed from a default FDD spectrum. Another is local spectral descriptors $\phi_{\ell }\in R^{224}$, obtained from a ±1 Hz neighbourhood around the target frequency $f_{0}$ across seven channels (first and second singular values, their ratio, derivatives, moving averages, and local variances), pooled into 32 segments per channel. The number of segments, 32, was empirically selected to balance spectral resolution and computational efficiency, providing sufficient detail to capture the modal peak shape while maintaining stable network training. The final feature vector is $\phi =\left[\phi_{g};\phi_{\ell }\right]\in R^{232}$. Since the input features are derived from the singular value spectra rather than the raw sensor channels, the trained model is independent of the number of measurement points. Hence, the model can be applied to structures with different numbers of sensors, provided that the same FDD preprocessing and feature extraction procedures are followed.

The network prediction is the softly weighted damping from the candidate damping estimates. The loss function, $L=\sum_{i=1}^{3}w_{i}{\left(\zeta_{pred}^{i}-\zeta_{true}^{i}\right)}^{2}$, was aggregated across modes using non-uniform weights. The first, second, and third modal damping ratios were assigned weights of 0.7, 0.2, and 0.1. Training is performed using the Adam optimiser with 1000 epochs.

Figure 7 illustrates that the loss function decreases steadily with training iterations, confirming the effectiveness of the model training. Figure 8 shows that the grid-weight model consistently improves damping ratio identification accuracy across all three modes. However, each identification requires extensive pre-computation of the candidate damping grid, which substantially increases the processing time and should be considered in practical applications. Based on timing over 10 repeated identifications, the runtimes were approximately 9 s for conventional EFDD, 51 s for step-wise EFDD, and 2544 s for grid-weight EFDD, respectively. Since the present model does not perform weighted aggregation on frequency estimates, its frequency identification results are identical to those of the conventional EFDD, and are therefore omitted here.

## 5. Experiments Application

### 5.1. Eight-Storey Steel Structure

Figure 9 presents the shaking table experiment of an eight-storey steel frame equipped with a negative stiffness damped outrigger (NSDO). The NSDO device functions to enlarge the damper displacement demand during seismic loading, thereby amplifying the damper’s hysteretic behaviour. Through this mechanism, it mitigates floor acceleration, inter-storey drift, and excessive deformation of the structure under earthquake excitation.

The tested specimen has a total height of 6 m with an aspect ratio of 5.6, and its main structural material is Q345 steel. The total mass is approximately $5.3\times 10^{3}$ kg. The NSDO was mounted at the fifth storey in conjunction with a viscous damper, whereas a weakened plate was incorporated at the first storey. Additional mass blocks were rigidly attached to avoid any unintended collisions with the frame plates during shaking. Eight accelerometers were positioned along the height of the column to measure absolute acceleration in the excitation direction. The sampling frequency was 256 Hz, and the white-noise excitation lasted for 300–305 s.

Table 2 summarises the shaking table input motions. WN denotes white noise, while SF, IV, and CC correspond to the San Fernando, Imperial Valley, and Chi-Chi earthquakes. Excitation intensity was gradually increased in six stages, each followed by a white-noise test, producing a total of 25 runs (see Figure 9). The experimental excitation was applied unidirectionally along the X-axis. The maximum displacement and acceleration in the Y-axis were less than 5% of those in the X-axis, indicating that torsional effects were negligible during the experiment. After the fourth round of testing (Run No.16), manual inspection revealed damage in the weak plates at the bottom of the core tube. At the end of the experiment, all weak plates exhibited residual deformation [63], as shown in Figure 10. Under excessive strain, the strain gauges detached (lost electrical continuity); accordingly, only manual inspection records are reported here.

The two proposed NN-embedded EFDD approaches were applied to the white-noise tests (WN1–7). Conventional EFDD with default parameter settings and the ERA-NExT method were adopted for comparison. The damping identification results are summarised in Table A1. The difference in natural frequency identification between the NN-embedded EFDD and the conventional EFDD was less than 0.01 Hz. Therefore, Table A2 only reports the results of EFDD-NN and ERA-NExT for comparison.

As shown in Figure 11 and Table A1, the damping ratio of the first mode exhibits a sharp increase at WN5, which is consistent with the manual inspection after Run No.16, where yielding of the weak plates was observed. In contrast, the second modal damping ratio shows a slight decrease at WN5, although the change is small and may fall within the range of identification uncertainty. The third mode does not present a systematic trend. The damping ratios identified by the NN-embedded EFDD are slightly larger than those obtained from the conventional EFDD. For all three modes, the natural frequencies decrease at WN5, which is physically reasonable since structural damage reduces stiffness and thus lowers modal frequencies. Since energy dissipation under ambient and seismic excitations is dominated by the fundamental mode, changes in damping are most evident in the first mode. A comparison between WN4 and WN5 shows that the damping ratio exhibits a relative change exceeding 20%, whereas the corresponding change in natural frequency remains below 2%. This highlights the higher sensitivity of damping ratios compared to frequencies in detecting minor structural damage.

### 5.2. Five-Storey Steel–Concrete Hybrid Structure

Figure 12 presents the shaking table experiment of a five-storey steel–concrete hybrid structure. The tested specimen has an overall height of 13 m, with each floor slab constructed of reinforced concrete measuring 5 m × 5 m. The applied loads, including self-weight, were 154 kN for the first four storeys and 136 kN for the fifth storey. The shaking table tests were conducted using the scale-up El Centro (1940) earthquake excitation record. The excitation inputs for the eight events are listed in Table 3. On each floor slab, two accelerometers were installed at the centre and at one corner to record three-directional acceleration responses. The sampling frequency was 256 Hz, and the white-noise excitation lasted 100–110 s. The NS direction was designated as the X-axis (grid 1–3 in Figure 12), and the EW direction as the Y-axis (grid A–D in Figure 12). A manual inspection was conducted after each event, with a detailed inspection carried out following the eighth event.

The corresponding peak ground accelerations (PGA) in the X-direction were approximately 0.04 g, 0.09 g, and 0.19 g for the 0.01, 0.25, and 0.50 intensity levels, respectively. In the Y-direction, the PGAs were slightly smaller, approximately 0.03 g, 0.07 g, and 0.14 g. The formation of visible cracks in the reinforced concrete columns was first observed during the fifth inspection. The extent and number of cracks increased progressively with successive excitation events, as shown in Figure 13.

The two NN-embedded EFDD approaches were applied to the white-noise response records in both the X and Y directions. The available record length (100–110 s) constrains the achievable spectral resolution to approximately Δf≥1/T≈0.01Hz. The step-wise model could not be used, as its WelchHead module includes a resolution option of 0.005 Hz, requiring a record length of at least 200 s to be physically feasible. The grid-weight EFDD-NN approach was employed, and, consistent with the previous section, the conventional EFDD and ERA-NExT methods were also applied for comparison. The identification results for damping ratios and natural frequencies are summarised in Table A3 and Table A4. The event numbering follows the configuration listed in Table 3. The first white-noise test conducted in Event 1 is designated as Event 0.

As shown in Figure 14, in the X direction, the grid-weight EFDD successfully captured the variation in the first-mode damping ratio at Event 5, while no consistent trend was observed in the results of the conventional EFDD or ERA-NExT. Between Events 0 and 4, the damping ratio identified by the grid-weight approach exhibited an upward trend. According to the data in Table A3, the damping ratio increased by approximately 34%, which is considerably higher than the 9% change observed in the corresponding natural frequency. This result highlights the superior sensitivity of the damping ratio in detecting subtle or visually imperceptible micro-cracks and minor damage in the structure. However, as the structural damage progressed further, the damping ratio decreased sharply, showing an opposite trend to its earlier behaviour. In contrast, the first-mode frequency continued to decrease after its abrupt drop at Event 5. Moreover, the second and third modal frequencies also demonstrated clear downward trends, consistent with the expected reduction in structural stiffness.

Figure 15 further indicates that the damping ratios identified in the Y direction exhibit no consistent correlation with structural damage. Although the identified frequencies contain a few local spikes, the overall trend for all three modes still gradually reduces. These observations suggest that the phenomenon may not be attributed to the accuracy of damping identification, but rather to the inherent limitations of using damping as a damage detection indicator.

### 5.3. Discussion of Damage Detection

The core objective of structural health monitoring (SHM) lies in damage detection. For invisible damage, as discussed in Section 5.1, the two proposed NN-embedded EFDD approaches show consistency with conventional identification methods. During WN3 and WN4, the identified damping ratios exhibited a noticeable increase, whereas the corresponding changes in frequency were negligible. This observation is consistent with the results in Section 5.2, where the grid-weight EFDD revealed similar behaviour in both the X and Y directions between Events 0 and 4. These findings collectively indicate that damping ratios possess higher sensitivity to subtle, non-visible damage. When minor internal damage occurs within structural components, the overall stiffness of the system remains nearly unchanged and thus cannot be reflected in frequency variations. However, micro-cracks, local yielding of steel, and the contribution of hysteretic energy dissipation increase the energy loss efficiency during vibration, which is manifested as an increase in the identified damping ratio.

In Section 5.1, when visibly detectable damage at WN5 occurred, all identification methods showed a sharp increase in damping ratio (approximately 20%), accompanied by a moderate decrease in frequency (around 2%). However, this advantage was no longer observed in the tests of Section 5.2. Only the grid-weight method presented a limited and consistent damping ratio trend in the X direction, while none of the methods showed any correlation between damping variation and structural damage in the Y direction. In contrast, frequency demonstrated a clear and stable decreasing trend. At Event 5, when minor cracking was first observed during inspection, the X-direction frequency exhibited a sharp reduction (approximately 37%). As the cracks propagated, the frequencies of all modes consistently decreased. Assuming that the grid-weight EFDD yields more reliable estimates, the damping ratios instead displayed a significant reduction, not an increase.

The inconsistency regarding the damage-detective capability of damping ratios can be attributed to three main factors. First, the white noise excitation for the five-storey structure lasted only 100 s, which was insufficient to ensure reliable damping identification accuracy. Second, the five-storey experiment represents a more complex configuration, as it included non-structural components such as bookshelves, office furniture, and computers. These elements introduced additional disturbance not only to the damping identification process but also contributed to fluctuations in the identified frequencies. Third, and most importantly, the eight-storey specimen is a pure steel structure, whereas the five-storey specimen is a steel–concrete hybrid structure. The uniform material properties of the steel frame result in a more stable energy dissipation mechanism, while in hybrid systems, multiple energy dissipation sources interact in a nonlinear and inconsistent manner. Previous studies have also shown that experiments exhibiting stable damping evolution with damage are typically associated with single-material structures [13,18,64] or individual components [12,14,15,16,20].

From a physical standpoint, the identified damping ratio from a vibration represents multiple energy-dissipation mechanisms, such as microcracking and crack closure in concrete, frictional slip at interfaces, joint and connection hysteresis, and local yielding. Incipient damage can alter the vibration energy loss per cycle without producing an immediate or commensurate change in global stiffness. Hence, damping behaves as a process-level indicator, whereas natural frequency is a stiffness-state indicator. In homogeneous, single-material systems, the dominant dissipation pathway tends to vary coherently with damage, and modal damping often exhibits a clear, monotonic shift at early stages when frequency remains nearly unchanged. In contrast, in hybrid or otherwise complex assemblies, multiple dissipation sources can be activated with different amplitudes and directions, producing competing effects (e.g., frictional increases vs. stiffness-loss-induced decreases in effective damping). These characteristics explain why damping can be markedly more sensitive to subtle or distributed damage in simple systems, yet less reliable or consistent in complex systems. Practically, damping-based indicators could therefore be interpreted as complementary to frequency. Their diagnostic value is maximised when excitation bandwidth is sufficient, record duration supports stable estimation, and mode-specific uncertainty is quantified. For heterogeneous systems, corroboration with stiffness-based metrics remains advisable.

### 5.4. Practical Implications for Simulation

In practical workflows, the proposed EFDD–NN modules can be used to calibrate and update damping models in numerical simulation and SHM pipelines. In finite-element analyses, the identified $\zeta_{n}$ under ambient excitation could offer damage- and time-dependent targets that replace fixed Rayleigh or uniform modal damping assumptions. For nonlinear or inelastic time–history analysis, bell-shaped proportional viscous damping models [56,65] can be informed by the identified evolution of $\zeta_{n}$ and $f_{n}$ in real structures, enabling a more faithful simulation of damage-dependent attenuation. In practice, coupling the learned $\left\{\zeta_{n}\left(t\right),f_{n}\left(t\right)\right\}$ with a bell-shaped formulation allows the damping envelope to reflect the observed progression of energy dissipation, which is expected to improve the accuracy of ambient demand predictions in time–history analyses.

## 6. Conclusions

This study proposed two neural-network-embedding strategies to improve the damping identification capability of EFDD for vibration-based structural health monitoring. The first approach (step-wise) decomposed the EFDD workflow into its bias-prone signal-processing steps. Neural networks were introduced as data-driven parameter selectors to replace default settings and reduce dependence on user experience. The second approach (grid-weight) relied on precomputing candidate damping ratios over a grid of parameter settings, followed by a neural weighting model that adaptively aggregated these candidates. Numerical simulations with additive measurement noise were employed to validate both strategies. The results demonstrated enhanced accuracy across modes. The grid-weight method achieves superior precision at the expense of higher computational cost due to the need for extensive pre-calculation.

The proposed models were further applied to two shaking table experiments, including an eight-storey steel frame and a five-storey steel–concrete hybrid structure. Compared with conventional methods, the grid-weight approach demonstrated superior consistency in damping identification and the ability to reveal damping variations before visible cracking, underscoring the potential of damping as a sensitive early-stage damage indicator. The experiments also highlighted the risks of damping-based damage detection. The energy dissipation mechanism is stable in homogeneous material structures, but not in hybrid systems with multiple energy-dissipation sources, such as friction, cracking, and interface slip, which introduce competing effects that obscure the relationship between damping and damage. Consequently, while the proposed grid-weight EFDD provides an improved and data-driven framework for damping identification, the interpretation of damping as a damage-sensitive parameter must still be considered material-dependent and complementary to frequency-based indicators.

## Figures and Tables

**Figure 1 sensors-25-06929-f001:**
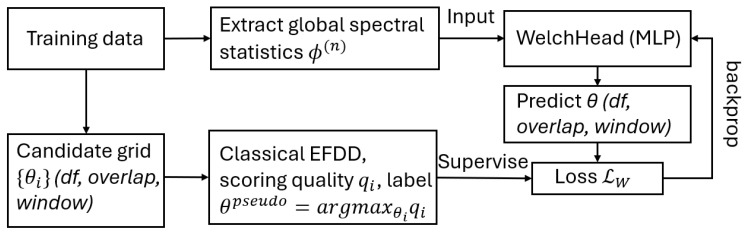
WelchHead workflow.

**Figure 2 sensors-25-06929-f002:**
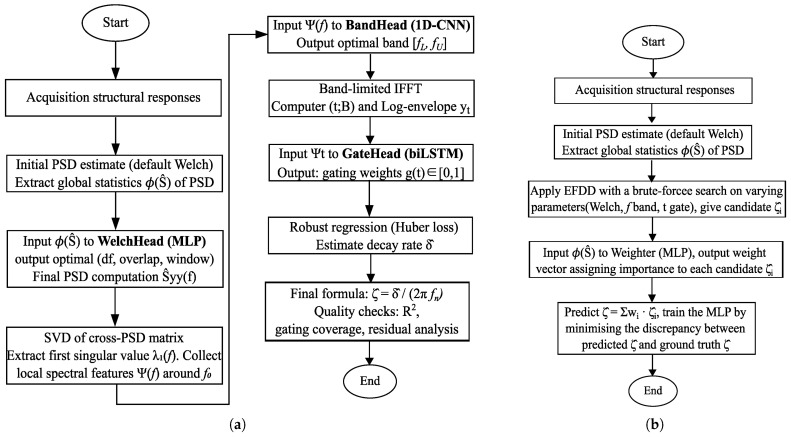
Flowchart of NN embedding on EFDD: (**a**) step-wise; (**b**) grid-weight.

**Figure 3 sensors-25-06929-f003:**
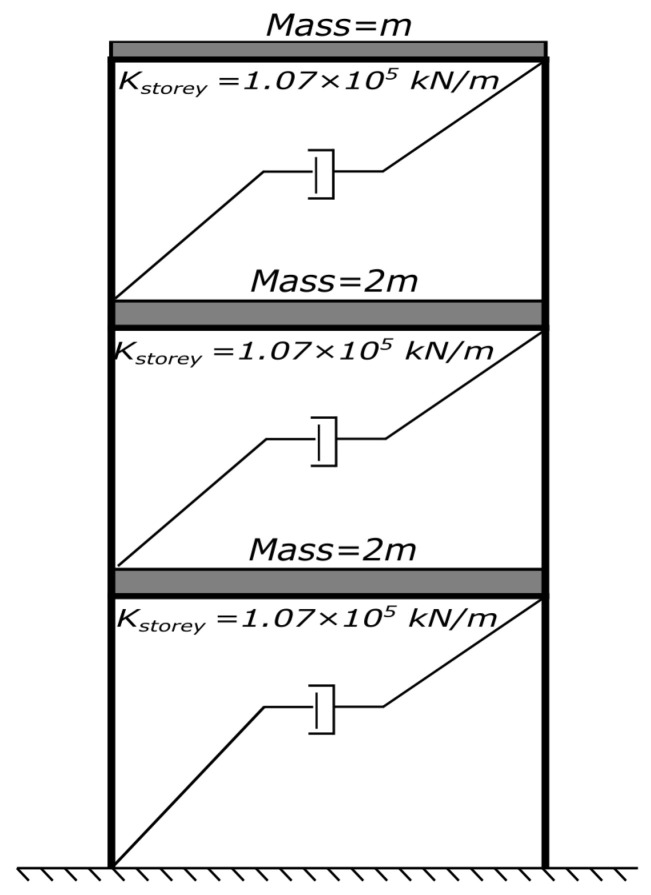
Schematic diagram of numerical model.

**Figure 4 sensors-25-06929-f004:**
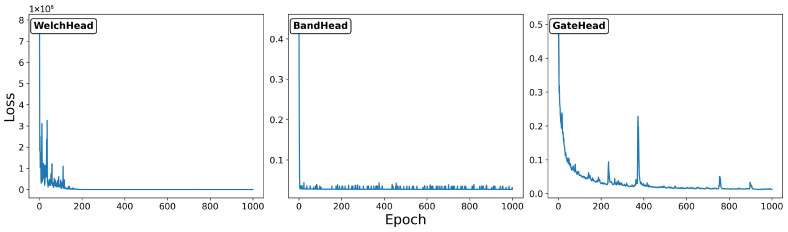
Loss function for step-wise training.

**Figure 5 sensors-25-06929-f005:**
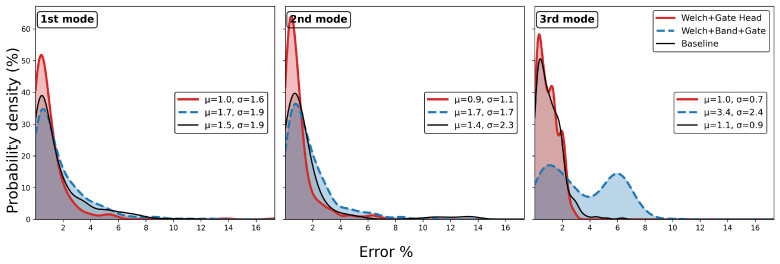
Test performance of step-wise vs. conventional module in frequency.

**Figure 6 sensors-25-06929-f006:**
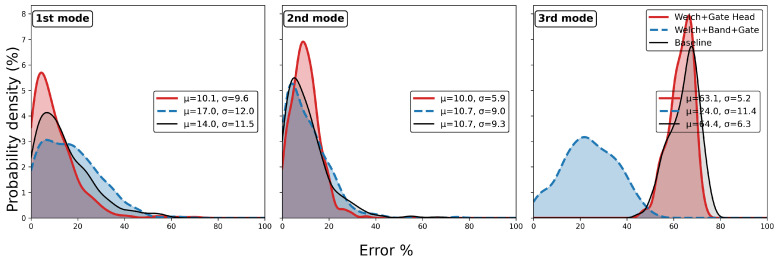
Test performance of step-wise vs. conventional module in damping.

**Figure 7 sensors-25-06929-f007:**
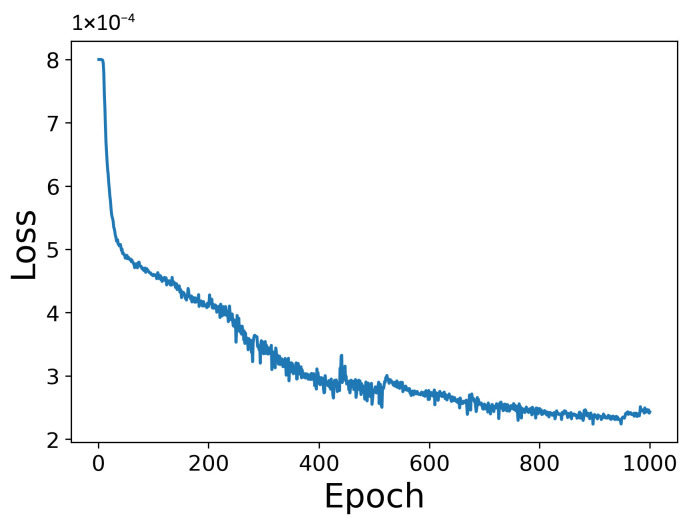
Loss function for grid-weight training.

**Figure 8 sensors-25-06929-f008:**
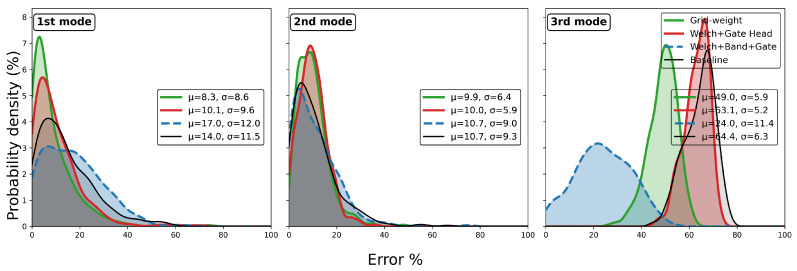
Test performance of grid-weight vs. conventional module on damping.

**Figure 9 sensors-25-06929-f009:**
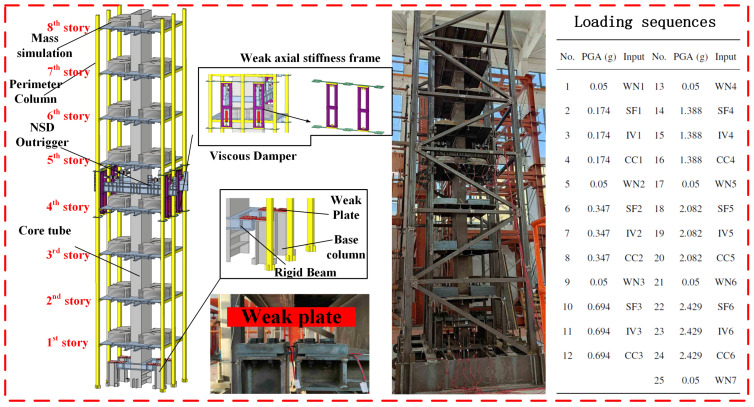
Overall view of the scaled NSDO structure and loading setup [63].

**Figure 10 sensors-25-06929-f010:**
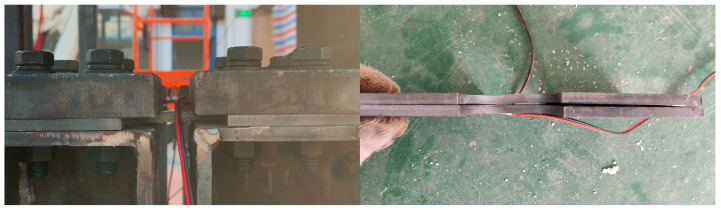
WeakPlate inspection.

**Figure 11 sensors-25-06929-f011:**
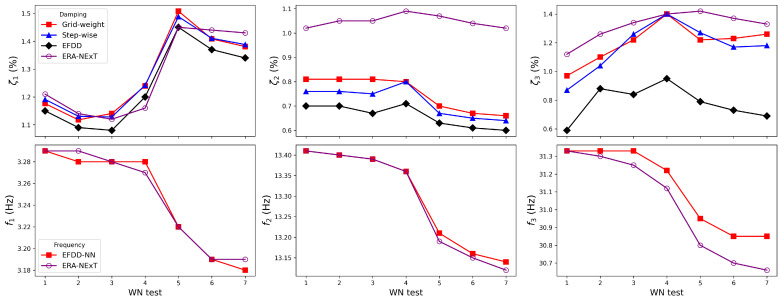
Identified damping and frequency by methods.

**Figure 12 sensors-25-06929-f012:**
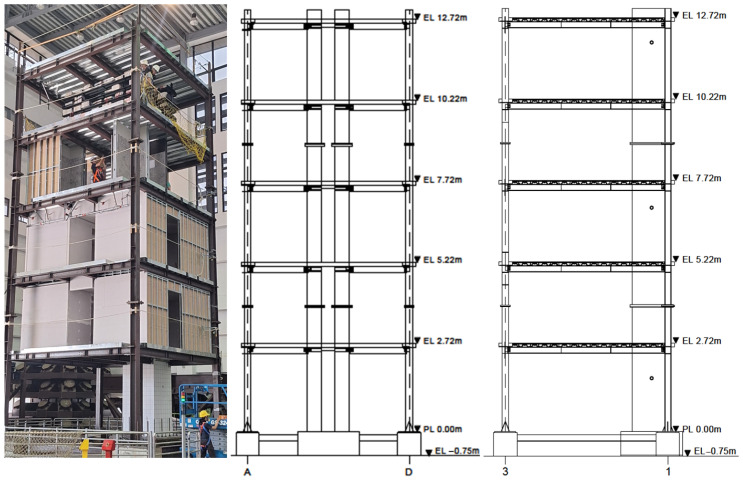
Overall view of the five-storey steel–concrete hybrid structure.

**Figure 13 sensors-25-06929-f013:**
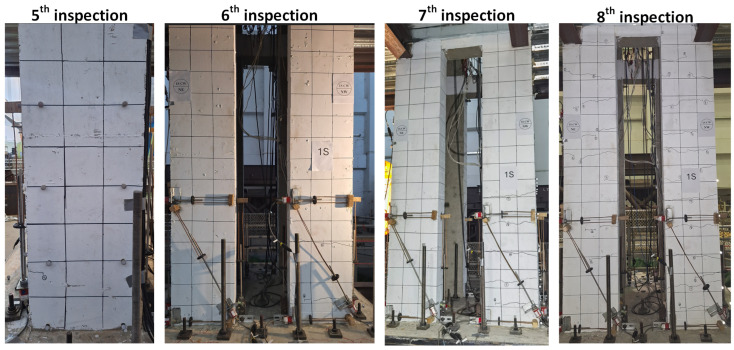
Photos after the fifth inspection.

**Figure 14 sensors-25-06929-f014:**
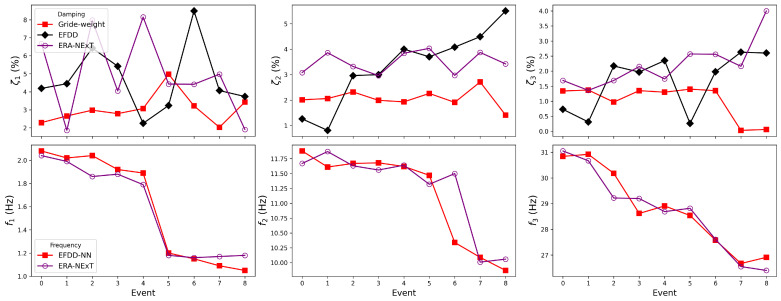
Identified damping and frequency by methods in the X direction.

**Figure 15 sensors-25-06929-f015:**
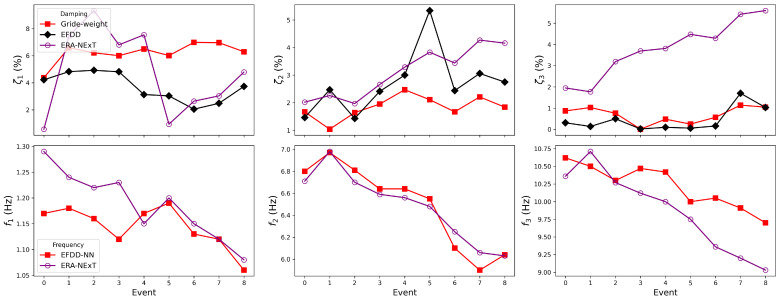
Identified damping and frequency by methods in the Y direction.

**Table 1 sensors-25-06929-t001:** Summary of the three proposed neural modules.

Module	Input	Output	Theoretical Objective	Training Strategy
WelchHead	Global spectral statistics $\phi (\hat{S})$	Welch parameters (df,overlap,window)	Maximise modal separation and minimise variance and spectral leakage	Pseudo-label imitation based on peak prominence and spectral smoothness scoring.
BandHead	Local spectral features $\left\{\Psi \left(f_{k}\right)\right\}$	Band boundaries $\left(\Delta f_{L},\;\Delta f_{U}\right)$	Balance linearity, monotonicity, bandwidth, contamination, and asymmetry	Pseudo-label imitation using envelope and spectral quality evaluation.
GateHead	Envelope features $\left\{\Psi_{t}\right\}$	Time-gating weights g(t)	Robust regression minimisation of Equation (Equation 16)	Pseudo-label imitation using offline optimisation of $\left(g^{★}\left(t\right),\;\delta^{★}\right)$.

**Table 2 sensors-25-06929-t002:** Information on input ground motions.

No.	Earthquake	Magnitude	Station	Year	PGA
1	San Fernando (SF)	6.61	Isabella Dam	1971	0.23 g
2	Imperial Valley (IV)	6.53	El Centro	1979	0.45 g
3	Chi-Chi (CC)	7.62	TCU084	1999	0.32 g

**Table 3 sensors-25-06929-t003:** Test events of five-storey steel–concrete hybrid structure.

Event	Intensity	Direction	G. Motion
1	0.01	-	White Noise
	0.25	EW	El Centro (1940) × 1.32
	0.01	-	White Noise
2	0.25	NS	El Centro (1940) × 1.32
	0.01	-	White Noise
3	0.25	EW + NS	El Centro (1940) × 1.32
	0.01	-	White Noise
4	0.25	EW + NS + UD	El Centro (1940) × 1.32
	0.01	-	White Noise
5	0.50	EW	El Centro (1940) × 1.32
	0.01	-	White Noise
6	0.50	NS	El Centro (1940) × 1.32
	0.01	-	White Noise
7	0.50	EW + NS	El Centro (1940) × 1.32
	0.01	-	White Noise
8	0.50	EW + NS + UD	El Centro (1940) × 1.32
	0.01	-	White Noise

## Data Availability

The data that support the findings of this study are available from the corresponding author upon reasonable request.

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
