# Peer review of "Improving EFDD with Neural Networks in Damping Identification for Structural Health Monitoring"

_sensors, 2025, doi:10.3390/s25226929_

Round 1

Reviewer 1 Report

Comments and Suggestions for Authors

1.The definition of “structural health monitoring (SHM)” is illustrated in the abstract,So it is no need to give the definition of “structural health monitoring (SHM)”  in the introduction (line 31)again.

2.it is need to provide the logic diagram of  WelchHead (MLP).

3.The installation positionof the sensors must be point out in the shaking table test in both steel frame equipped with a negative stiffness damped outrigger and steel–concrete hy-brid structure

4.What is the damage patternof the shaking table test specimens, is there any photonscan be provide?

Author Response

We sincerely thank the reviewers for their careful reading and constructive suggestions. The manuscript has been revised accordingly. In each modified passage, the revised parts are highlighted, and the line numbers are provided to delimit the full edit range.

Below, we respond to comments one by one:

Comment 1 – “The definition of “structural health monitoring (SHM)” is illustrated in the abstract,So it is no need to give the definition of “structural health monitoring (SHM)”  in the introduction (line 31)again.”

Response: Thank you for the helpful observation. While the Abstract offers a concise overview, the Introduction is often consulted independently and sets the scope and terminology for the main text. To support readers who may not read the Abstract in parallel, a single, short SHM definition is retained in the Introduction. The wording has been kept brief to avoid redundancy while preserving clarity for a broad audience.

Comment 2 – “it is need to provide the logic diagram of  WelchHead (MLP).”

Response: A logic diagram illustrating the WelchHead (MLP) workflow has been added; see Figure 1

Comment 3 – “The installation position of the sensors must be point out in the shaking table test in both steel frame equipped with a negative stiffness damped outrigger and steel–concrete hybrid structure”

Response: Sensor layouts and installation positions for the steel frame with a negative-stiffness damped outrigger and for the steel–concrete hybrid specimen have been added in the experimental description (lines 384–387 and lines 425-428).

Comment 4 – “What is the damage patternof the shaking table test specimens, is there any photonscan be provide?”

Response: The observed damage patterns are illustrated in Figure 10, with corresponding descriptive text at lines 397–398.

Reviewer 2 Report

Comments and Suggestions for Authors

The manuscript presents an improved EFDD by integrating neural networks, replacing heuristic parameter choices with data-driven modules. To tackle the intrinsic deficiencies in damping identification due to bias introduced at several signal-processing stages. The proposed strategies were validated through numerical simulations and shaking table tests.

The manuscript concept is very interesting, the writing and figures are clear, the analysis is adequate, and the conclusions are supported by the results.

I highly recommend accepting the manuscript considering the following comments:

  • I suggest adding a sub-section for practical applications
  • In line 379, please elaborate more on “the damping ratio of the first mode exhibits a sharp increase at WN5, which is consistent with the manual inspection after Run No.16, where yielding of the weak plates was observed.”
  • Is the method affected by the structural material such as masonry, wood, concrete?
  • Can the results be extended to bridges?

Considering the above points will enrich the paper.

Thank you for considering the above comments.

Finally, I want to admire the great effort and figures provided.

Author Response

We sincerely thank the reviewers for their careful reading and constructive suggestions. The manuscript has been revised accordingly. In each modified passage, the revised parts are highlighted, and the line numbers are provided to delimit the full edit range.

Below, we respond to comments one by one:

Comment 1 – “I suggest adding a sub-section for practical applications.”

Response: Thank you for the constructive suggestion. A concise subsection entitled “Practical implications for simulation” has therefore been added at lines 524–533, outlining how the identified, mode-specific damping ratios can calibrate proportional damping in finite-element analyses to improve response prediction with damage-dependent attenuation. At this stage, a dedicated field-scale case study is not included because staged damage records suitable for a verifiable demonstration are not available. The manuscript thus remains validated by two controlled shaking-table experiments, and incorporation of well-documented field datasets is planned when appropriate data become accessible.

Comment 2 – “In line 379, please elaborate more on ‘the damping ratio of the first mode exhibits a sharp increase at WN5, which is consistent with the manual inspection after Run No.16, where yielding of the weak plates was observed.’”

Response: Additional description and visual evidence have been provided in Figure 10 with explanatory text at lines 397–398, clarifying the observed yielding of the Weakplates.

Comment 3 – “Is the method affected by the structural material such as masonry, wood, concrete?”

Comment 4 – “Can the results be extended to bridges?”

Response (for 3 & 4): A concise statement has been added in the Introduction (lines 61–66) clarifying that EFDD, as an output-only frequency-domain method, is largely material- and typology-agnostic. The proposed neural modules do not alter EFDD’s core assumptions and are therefore compatible with buildings (masonry, timber, reinforced concrete) and extendable to bridges under ambient and operational excitation, subject to approximate linearity of response.

Reviewer 3 Report

Comments and Suggestions for Authors

The paper presents an innovative hybrid approach integrating neural networks with the Enhanced Frequency Domain Decomposition (EFDD) method for damping identification in structural health monitoring. The study is technically solid and promising; however, several points require clarification and refinement before publication:

  1. Clarify the neural network training process like pseudo-labels or “quality scores” for the Step-wise models and describe measures taken to prevent overfitting and ensure generalization.
  2. Add key experimental details including the sampling rate, test duration, and sensor layout for both shaking-table experiments, and etc.
  3. Expand discussion on damping sensitivity.
  4. Report computational aspects of approximate runtime or computational cost comparison between Grid-weight EFDD, Step-wise EFDD, and conventional EFDD.
  5. Add 2022–2024 literature on neural-network-based or ML-enhanced EFDD methods.

Author Response

We sincerely thank the reviewers for your careful reading and constructive suggestions. The manuscript has been revised accordingly. In each modified passage, the revised parts are highlighted, and the line numbers are provided to delimit the full edit range.

Below, we respond to comments one by one:

Comment 1 – “Clarify the neural network training process like pseudo-labels or “quality scores” for the Step-wise models and describe measures taken to prevent overfitting and ensure generalization.”

Response: The pseudo-label and quality-score construction for the Step-wise model, together with safeguards against overfitting and generalisation protocols (early stopping, micro-checks), are now described at lines 137–140 and lines 212–217.

Comment 2 – “add key experimental details including the sampling rate, test duration, and sensor layout for both shaking-table experiments, and etc.”

Response: Sampling rate, excitation duration, and sensor layout details for both shaking-table experiments have been added at lines 384–387 and lines 425-428.

Comment 3 – “Expand discussion on damping sensitivity.”

Response: A dedicated paragraph discussing damping sensitivity—its physical basis, contrast with frequency shifts, and differences between simple and complex assemblies—has been added at lines 505–522.

Comment 4 – “Report computational aspects of approximate runtime or computational cost comparison between Grid-weight EFDD, Step-wise EFDD, and conventional EFDD.”

Response: A concise runtime comparison has been included at lines 368–370, reporting runtimes over repeated runs for the three methods.

Comment 5 – “Add 2022–2024 literature on neural-network-based or ML-enhanced EFDD methods.”

Response: Recent literature from 2022–2024 has been incorporated in the Introduction at lines 70–74, with corresponding entries added to the References (Refs. 35–37), situating the present contribution within current ML-assisted frequency-domain identification research.